# Investigation on Musculoskeletal Injury and Psychological Empowerment of Reflexologists in Taiwan: Analysis of the Recognition to Alternative Therapy

**DOI:** 10.3390/healthcare11030394

**Published:** 2023-01-30

**Authors:** Ching-Yun Chen, Deng-Chuan Cai

**Affiliations:** Graduate School of Design, National Yunlin University of Science & Technology, Douliu City 640301, Taiwan

**Keywords:** complementary and alternative medicine, foot massage, musculoskeletal strain, job cognition, massage tools

## Abstract

Many studies have proven that reflexology has been used as a complementary medical treatment. Therefore, the government has started to plan an examination system for reflexology personnel to ensure the quality of service. Reflexologists work long hours, have heavy workloads, and perform poses that do not conform to human factors, which often cause musculoskeletal fatigue. The purpose of this study is to understand the musculoskeletal pain conditions of reflexologists, the psychological empowerment status, and the perceptions of complementary medicine therapy. The data for this study were obtained in two ways: (1) 59 practitioners were surveyed by using a face-to-face questionnaire and (2) a semi-structured interview was carried out for 10 practitioners. This study discovered the following: (1) Reflexology practitioners have musculoskeletal discomfort symptoms in body parts, including the left shoulder (25.4%), left hand or wrist (25.4%), lower back (25.4%), right shoulder (23.7%), left elbow or forearm (22%). (2) Reflexology practitioners are highly psychologically empowered to work. (3) The practitioners of foot therapy hold a positive attitude towards foot therapy and believe that foot therapy is a natural therapy, which is self-serving and can help others. (4) Most reflexologists support the government’s desire to promote the reflexology examination system and are willing to help develop the policy. (5) The height of most reflexologist work chairs does not match the height of the guest’s seat and is not ergonomic.

## 1. Introduction

Fr. Josef’s method of foot reflexology (FJM hereinafter referred to as foot therapy) has various mechanisms:Alleviates abnormal organ and gland function: After applying pressure, pushing, and rubbing each reflex area of the foot, this area will generate a nerve pathway connecting the corresponding organs, glands, and systems. Pressing the reflex area of the foot can gradually eliminate the pain points of sand-like reactants in the foot and promote abnormal organs, glands, and systems, alleviating symptoms or pain. The plantar reflex area corresponds to the organ it belongs to.Circulation: Circulation helps vein and lymph flow; restores blood circulation, removes calcium deposits, uric acid, and crystals; regulates the body; eliminates fatigue; and increases energy [1,2].Balance effect: Massage affects the nervous, motor, and cardiovascular systems, allowing the entire body to relax, take deep breaths, and reduce fatigue [3].Pain relief: The reflex area of the foot is pressed to generate electrical pulse waves. The gate theory states that stimulating A-beta nerve fibers inhibits pain-conducting C nerve fibers and releases inhibitory neurotransmitters, causing the calcium channel that transmits pain to the brain to close, thus preventing pain from reaching brain consciousness.

Most studies have shown that foot therapy can reduce all kinds of pain, including migraine [4], abdominal pain [5], colic [6], low back pain [7], joint pain [8], pain from organ removal or transplantation [9,10,11], various cancer pain [12], and medical examination pain [13].

Foot therapy has been used to treat isolated musculoskeletal cases [14] and insomnia [15,16].

Foot therapy is effective in patients with diabetes, premenstrual syndrome, cancer, multiple sclerosis, idiopathic detrusor hyperactivity, and dementia and anxiety in patients with coronary artery disease [17,18].

The founder of Taiwan foot therapy, the Swiss missionary Fr. Jose Eugster, was suffering from a family joint disease. He developed foot therapy to cure his knee disease after being introduced by Brother Xue to the book *Future Health* by Swiss nurse Martha Weiwen. In 1980, Fr. Jose Eugster’s foot therapy method was established, engaged in empirical research and continuous improvement, and promoted throughout Taiwan [19]. During the early promotion period of Fr. Jose Eugster, the name of the method was undetermined. Therefore, many people opened shops and filed for registration under the name Fr. Jose Eugster, and various foot treatments were developed.

According to Taiwan’s labor insurance occupational disease benefits, musculoskeletal injuries are the most common injuries. In terms of occupational low back pain alone, there were 120 cases in 2019, 128 cases in 2020, and 91 cases in 2021. As for arm, neck, and shoulder diseases, there were 353 cases in 2019, 360 cases in 2020, and 356 cases in 2021 [20], showing that related diseases still cause great harm to workers. Long-term sitting working postures used during foot massages have been identified as high-risk working postures [21]. According to relevant surveys in Taiwan, the top three symptoms of musculoskeletal injuries and discomforts among massage practitioners include shoulders (65.0%), lower back (61.7%), and wrists (57.1%) [22]. The hazard risk of musculoskeletal injuries among practitioners in foot massage establishments in central Taiwan shows the highest frequencies of injuries in the lower back (10.83%), upper back (9.94%), and left hip and thigh (7.86%) [23]. Musculoskeletal injuries reduce labor income, with a concomitant increase in the annual amount of labor compensation and medical expenses. The relative social and economic burden caused by musculoskeletal injuries far exceeds that of other occupational diseases. Therefore, the problem of musculoskeletal injuries should not be underestimated. Poor posture, workstation design, prolonged static trunk and neck flexion, excessive trunk twisting, unnatural static posture, and high stress and shear forces on the spine are the leading causes of musculoskeletal disorders among massage therapists [24]. In addition, personal factors (age or gender), organizational and physical factors (working hours, repetitive tasks, and stress), and psychosocial factors affect the incidence of musculoskeletal disorders [25].

Empowerment is defined as the concept that people have the ability to understand and control themselves and their environment (including the social, economic, and political spheres), enhancing their capabilities and prospects and exhibiting higher levels of achievement and satisfaction [26]. Empowerment is an act that enables a person to take steps on their own to achieve a goal that they set for themselves. Empowered individuals experience high levels of self-esteem, self-efficacy, and control over their activities [27]. Psychological empowerment requires an assessment of work meaning, work autonomy, work ability, and work impact. Work meaning represents the value of the task and its goals, it is related to personal standards and value systems, and it reflects the individual’s overall interest in the task [28]. Work autonomy is an individual’s sense of independence in taking initiative and controlling work and expresses the degree of self-determination in work behavior and process [29]. Work competence is the degree to which employees perceive themselves as able to perform tasks with skill [30], which has a positive impacts on performance [31]. Job influence can improve job motivation [32].

Taiwan’s massage industry gradually evolved from an exclusive industry to a tourism and cultural industry. Before 2010, the massage industry was exclusively for the visually impaired. In 2011, people with clear vision were allowed to engage in massage [33]. In 2012, the government took the massage industry seriously and included it in the cultural, sports, leisure, and other service industries of the Ministry of Health and Welfare of the Examination Yuan government and developed certification and masseuse training programs. In Taiwan, the massage industry is a characteristic culture of the tourism and leisure industry. It can develop the tourism industry and increase its economic output value. The impact of the epidemic significantly changed the market structure, consumption profile, and travel trends of tourists coming to Taiwan in 2020 compared to those in previous years. From January to March 2020 (before the border strict control during the epidemic), massage and shiatsu accounted for 9.52%, ranking No. 8; July–December 2020 (after the relaxation of epidemic border control) accounted for 14.08%, rising to No. 3 among reflexology practices [34]. Shiatsu, massage and foot massage, or meridian massage are well recognized and loved by Chinese people and international tourists.

Currently, 1159 operators in Taiwan are registered in the “foot massage industry,” with over 60,000 registered massage service personnel [35,36]. Kuan-Chia Lin et al.’s research revealed that the one-year utilization rate of complementary and alternative medicine in Taiwan was 85.65% [37]. Few studies in Taiwan have discussed the work cognition, job satisfaction, strain injury status, and practical experience of reflexologists. Therefore, this study aimed to investigate job cognition, job satisfaction, and the current status of work environment and equipment among reflexology practitioners in Taiwan.

## 2. Materials and Methods

This study was approved by the National Cheng Kung University Human Research Ethics Review (No. 109–584). A questionnaire survey was conducted after obtaining informed consent signed by the foot reflexology therapist.

The research tool is based on Lin et al. [37] and Bausell et al. [38]. Documentation and the Chinese translation of the Nordic Musculoskeletal Questionnaire (NMQ) developed by the Occupational Safety and Health Administration of the Ministry of Labor of Taiwan [39] were compiled, and the first draft of the questionnaire on musculoskeletal injuries and psychological empowerment of Taiwanese reflexologists was completed. Finally, the following individuals were invited: Ms. Huang, head nurse of the medical service industry; Dr. Han, an engineering expert in electromechanical and electrical engineering; (3) lecturers Mr. Chen, Mr. Jiang, and Mr. Yan in the clinical reflexology system; lecturers Ms. Wang and Ms. Lin worked on the validity of the questionnaire and then compiled and revised review opinions to complete the approved version of the questionnaire, with a Cronbach’s alpha of 0.84.

This study utilized two questionnaires and interviews. The questionnaire adopted face-to-face deliberate sampling to investigate the working conditions and musculoskeletal pain of reflexologists in Taiwan. In total, 60 questionnaires were distributed face to face to stores registered for foot reflexology massage in the four regions, including northern, central, southern, and eastern Taiwan; among these, 59 questionnaires were returned, with a recovery rate of 98.3%. Thus, to investigate the working conditions and musculoskeletal pain problems of foot reflexology therapists in Taiwan, 59 reflexologists (31 males and 28 females) were investigated using a questionnaire.

Five questionnaires were utilized, including (1) basic information, (2) living conditions, (3) working conditions, (4) musculoskeletal pain scale, and (5) psychological empowerment scale [37]. The musculoskeletal pain scale is a modified Nordic musculoskeletal pain scale, with checklists commonly used by Taiwanese workers [39]. This study adopted a 0–5 point scale, with 0 indicating no pain and free movement of the joints. One indicates slight pain and joint movement at the limit will be sore but can be ignored. Two indicates moderate pain, and the joint will be sore when more than half of the movement is performed, but the full range of motion can be completed, which may affect work. Three indicates severe pain, and joint activity is only half that of a normal person, affecting work. Four indicates very severe pain, and joint activity is only 1/4 that of a normal person, affecting voluntary movement ability. Finally, five indicates extreme pain, and the body cannot move independently. Figure 1 depicts the musculoskeletal pain scale. The psychological empowerment scale (PES) is a psychological cognitive questionnaire that measures the degree of psychological empowerment of individuals in the workplace [40]. The psychological empowerment scale is a 5-point psychological cognitive questionnaire (1: strongly disagree; 5: strongly agree). The higher the score, the greater the psychological empowerment. There were 59 survey respondents for the 5 questionnaires.

In total, ten reflexology practitioners were interviewed for this study using a semi-structured questionnaire developed by two experts (professor Lin, a professor at the University, and lecturer Mr. Weng, a plantar reflexologist). In terms of the current lack of knowledge of foot reflexology practitioners on foot therapy, this included physical symptom attributes, techniques, tools, and whether the podiatry practitioners know that the government started gathering relevant data on the examination system. One is the founder of reflexology (Fr. Josef Eugster), two were local aborigines (the foot reflexology therapists who followed Fr. Josef Eugster in the early days), three were Taiwanese residents, and four were new residents. The following is the interview outline: (1) What is your opinion on foot massage? (2) Which client gives you the highest sense of accomplishment from your reflexology client base? (3) Do you use massage devices at work? (4) What is your opinion on the government’s implementation of the examination for foot reflexology practitioners in the future? The interview outline is as follows: A qualitative design was adopted, and 10 people were interviewed using a semi-structured questionnaire to summarize reflexologists’ views on their work. The data were analyzed with MAXQDA software code to summarize the opinions of the practitioners of foot reflexology.

The research period was from 17 January 2022 to 5 July 2022. Interviews were conducted in Chinese and Taiwanese language. Subsequently, data were counted using SPSS V22.( International Business Machines Corporation , New York, United States)

## 3. Results

### 3.1. Basic Information

This study included 59 reflexologists: 31 were male and 28 were female. Table 1 presents the basic information of respondents. In this survey, the proportion of male and female respondents is similar (52.5%, 47.5%), with an average of 49 years old, placing them in the middle age group. The age distribution of men and women is shown in Figure 2. The average body mass index (BMI) of 59 reflexologists was 25.1, placing them in the overweight category (24 ≤ BMI < 27). All 59 participants stated that they received foot massage training. Among the 59 employees, 26 were engaged in pedicure work from the beginning, and 33 employees were engaged in business (32.2%), labor (11.9%), clerk (3.4%), agriculture (5.1%), and other (3.4%) careers before changing jobs. Twenty-one people received other occupational training before they were exposed to reflexologist training (Table 2).

### 3.2. Living Conditions

The questionnaire employed included the following questions:

1. Do you have any health problems or disabilities?

The survey revealed that 76.3% (45) were disease-free, while 23.7% (14 respondents) of the respondents had chronic diseases, including eye, high blood pressure, heart, bronchial, cervical spine pain, and ear diseases (Table 2).

2. Which part of your body has been injured before?

Thirty-five (59.3%) respondents had no trauma, while 40.7% (24 respondents) had musculoskeletal injuries, including head, spine, and limbs (Table 2).

3. Are you currently conscious of your health status?

The highest possible score is 5 points for conscious physical health, the lowest is 1 point, and the average satisfaction score is 3.9 points. This shows that most reflexologists are in good health (Table 1), while in this study, only 8.5% were unhealthy, and 67.8% were healthy (Table 2).

4. Do you think you are living a happy life now?

The highest possible score is 5 points, the lowest is 1 point for the current living situation, and the average satisfaction score is 4.0 points. This implies that most reflexologists are content with their living conditions, indicating that the level of happiness in life is higher than the level of health (Table 1).

5. What is your current job satisfaction?

The highest possible score for job satisfaction is 5 points, the lowest is 1 point, and the average satisfaction rate is 4.1 points. This shows that most reflexologists have high job satisfaction, indicating that job satisfaction is better than health (Table 1).

6. Do you currently have regular leisure activities?

The survey revealed that reflexologists value leisure life; 57.6% (34) regularly engage in leisure activities, with low-intensity walking (25.4%) being the most popular. Additionally, it was discovered that 10.2% (6) of the reflexology practitioners had regular volunteer services. (Table 2)

### 3.3. Working Conditions

1. What is the nature of your current job: □ Full-time □ Part-time □ Trainee?

It was discovered that 47 respondents (79.7%) practiced reflexology full-time. In contrast, part-time workers (12 respondents, 20.3%) are engaged in full-time work during the day and use evenings or holiday appointments to serve customers (Table 2).

2. How long have you been working in the reflexology industry?

The shortest period is 1 month, the longest is 36 years, and the average is 9.6 years (SD = 9.5) (Table 1). This implies that reflexology is a viable career option.

3. What is your current workplace in the reflexology industry (multiple choices): □ work platform □ to the house □ studio □ Others

In total, 89.8% of reflexologists work on a work platform, 5.1% provide in-house service, 3.4% adopt mixed methods, and 1.7% work in studios. This shows that most places where reflexology work is performed have fixed storefronts (Table 2).

4. How many days do you currently work per week?

The maximum is 7 days per week, the minimum is 0.5 h, and the average number of days per week is 5.2 (Table 1).

5. How many working hours do you engage in reflexology every day on average?

The maximum service hour per person daily is 12, the minimum is 0.1, and the average is 7.3 (Table 1).

6. How many customers do you serve on average per day?

The maximum number of customers served daily is 10, the minimum is 1, and the average is 3.8 (Table 1). Unless the customer specifically designated reflexology service staff, respondents in eastern Taiwan served customers randomly on the work platform in a shift arrangement. Respondents in western Taiwan mostly serve customers via a pre-booking system.

7. Overall, do you love your job? □ very dislike □ not very fond of □ still love □ love □ very loving

The maximum value of the degree of love for work is 5, the minimum value is 1, and the average is 4.1, showing that reflexologists enjoy their job (Table 1).

8. Overall, would you be willing to spend time privately learning new technologies? □ very unwilling □ reluctant □ fairly willing □ willing □ very willing to

The maximum value is 5, the minimum value is 1, and the average score is 4.0, indicating that reflexologists are willing to learn new technologies (Table 1). Furthermore, it was discovered that a reflexology school requires qualified reflexologists to participate in on-the-job training and training every year and share new findings with the association. In addition, interviewees from the faction’s eastern work platform stated that every Tuesday evening, they would gather to discuss difficult problems encountered at work and exchange ideas. New reflection areas will be reported to the association for empirical research if they are discovered.

9. Overall, are you satisfied with your work environment? □ Very dissatisfied □ Not so satisfied □ Pretty satisfied □ Satisfied □ very satisfied

Work environment satisfaction is 5 points, the minimum value is 1 point, and the average is 4.0 points. This shows that most respondents are highly satisfied with their work environment (Table 1).

10. Overall, what is the biggest trouble for you in your current job? □ No □ Yes

In total, 34 (57.6%) reflexologists have no problems with their current work, and 42.4% (25) reflexologists have the following problems: 13.6% (8) are affected by the epidemic, 8.5% (5) have personal problems, 6.8% (4) have questions from guests, and 13.6% (8) have work problems. The reasons are that the epidemic affects the number of customers, promotion is difficult, transportation time to and from the workplace is long, customers are numerous, and there are insufficient human resources (personal studio). Work and rest are abnormal, hand size is small and makes exerting forces difficult, and the customers are concerned about service times and other problems (Table 2).

11. Do you do some self-protection exercises before work every day?

In total, 79.7% of reflexologists will perform self-protection exercises, such as deep breathing, mutual massage, and warm-up exercises before work, attaching importance to basic work protection (Table 2).

12. Do you use devices at work?

The massage stick is used by 59.3% of reflexologists to assist respondents in applying force to the reaction area. However, 40.7% of reflexologists applied force with bare hands, without gloves, and using small tools (Table 2).

13. What kind of customers do you serve most?

The most frequently served customers are those with chronic diseases, and the top three include insomnia, muscle ache problems, and gastrointestinal problems (Figure 3).

14. Hardware size

Reflexologist’s chair height Centimeter □ With designed chair □ Others

Served Seat Height Centimeter □ With designed chair □ Others

Served person poses □ Sitting □ Lying □ Others

In total, 57.6% of the working chairs of reflexologists are fixed and do not have backrests and pulleys; 42.4% have height-adjustable seats, backrests, and pulley facilities. The height range of the work chair is 20–90 cm. The height range of the guest seat is 30–120 cm. The height of most work chairs does not match the height of the guest’s seat and is not ergonomic. (Table 2).

### 3.4. Investigation of Musculoskeletal Symptoms

Statistics revealed that 33.9% (20 respondents) (Table 2) of reflexologists had musculoskeletal discomfort in more than one body part in the past year, including the left shoulder (25.4%), left hand or wrist (25.4%), lower back (25.4%), right shoulder (23.7%), and left elbow or forearm (22%), and these comprise the top five most uncomfortable body parts. Moreover, the average pain level ranged from none to mild pain (0.2–0.6). The data revealed few symptoms of discomfort (Table 3).

### 3.5. Questionnaire on Work Attitude

The psychological empowerment scale (PES) mainly evaluates four aspects of cognition, including work meaning (meaning), autonomy (self-determination), ability (competence), and work impact (impact) [40].

Work meaning includes three sub-tasks: the meaning of the work, the meaning to the individual, and the importance of the work. Autonomy encompasses an individual’s ability to make work-related decisions, independence, and an assessment of autonomy. The competency aspect covers job mastery skills, self-confidence, and ability confidence. Work influence refers to the degree of individual’s influence, control, and influence on the operation of work organization. Reflexology work was meaningful to the respondents, with an average of 4.4, 4.5, and 4.2, respectively, implying strong support for the reflexology profession.

Respondents’ autonomy in reflexology was 4.4, 4.4, and 4.3, respectively, showing that reflexologists have a high level of autonomy in their work.

The respondents’ self-efficacy assessment of reflexology work was 4.5, 4.5, and 4.5, respectively, implying that reflexologists have a high confidence level in completing the job.

The average number of respondent influence on the organization of reflexology work is 3.2, 3.2, and 3.1, respectively.

Among the four dimensions, work meaning and self-efficacy have the highest scores, while influence dimensions have the lowest scores.

The top four items with the highest scores are as follows: “2. What I do at work is very meaningful to me personally” in the meaning dimension (M = 4.5, SD = 0.7). In the talent dimension, “7. I have mastered the skills needed to complete the work” (M = 4.5, SD = 0.6), followed by “8. I am confident that I can do all things well at work” (M = 4.5, SD = 0.6) and “9. I am very confident in my ability to complete the work” (M = 4.5, SD = 0.6). The top three items with the lowest scores were as follows: “10. I have a great influence on what happened in this department” (M = 3.2, SD = 1.1), and “11. I have a great influence on what happened in this department” in the impact dimension, followed by “This is a great control” (M = 3.2, SD = 1.1) and “12. I have a significant influence on what happens in this department” (M = 3.1, SD = 1.1) (Table 4).

### 3.6. Interview Part

1. What is your opinion on foot reflexology?

“Very good therapy, because one can get healthier” (A).

“I believe that God had placed the best health law upon us when he made us” (C).

“It is a good health law, not only for your health, but also to help others, and most importantly, to make money” (E).

“Reflexology is worth learning for everyone and can make everyone truly healthy” (I).

In summary, reflexology practitioners believe that reflexology can improve health (100%), help people (60%), and increase economic income (20%) (Table 5).

“Do not talk about foot massage; talk about foot reflexology” (E).

“I think foot massage is more of a business model. Foot therapy is different from foot massage. Previously, when Fr. Josef Eugster visited Taiwan for the first time, there was a language, cognition, and communication gap. In order to preach to the people, he did not refute. However, the foot therapy FJM is now officially registered with the government. Therefore, Fr. Josef Eugster will try his best to clarify the name of our foot reflexology health method” (D).

2. Who has been your most sense of accomplishment guest?

“The brother of a client who was in a coma after a stroke called the association for help, and I taught him to do foot reflexology for his brother every day. As a result, he regained consciousness and continued to do it. The client is currently as free as an ordinary person (recovered like a normal person)” (Aboriginal B).

“After foot reflexology, a client with stroke and gout now lives in Lanyu and dances happily every day as if everything is fine” (New Resident E).

“Forty years ago, Li Wen, the host of the police radio station, had a goiter. After a foot reflexology, he went back to Taipei for a check-up and was fine (the test result was normal)” (Swiss C).

“I am in charge of special conditions and rare diseases, such as serving the clergy to heal and re-energize their bodies” (Taiwanese D).

The clients of reflexologists are all patients with chronic diseases, with the most being neurological problems (60%). A session with a reflexology practitioner improves the disease (100%) and assists in curing the disease (50%) (Table 5). The most profound cases include the following: gradually improving (Aboriginal A), the recovery of consciousness and mobility after stroke (Aboriginal B), cure of goiter (Taiwanese Swiss C), clergy psychosomatic disorder (Taiwanese D), stroke and gout cured (new resident E), spleen enlargement is cured (new resident F), insomniacs do not need to rely on drugs to sleep (new resident G), early detection of the client’s physical condition (new resident H), chorea will not worsen (Taiwanese I), and stroke and paralysis are cured (Taiwanese J).

3. At work, do you use reflexology tools?

“I always use a reflexology stick because it is a basic tool, it saves effort and gets to the deeper muscles” (E).

“We all use reflexology sticks with our hands, and I think it is effective” (A).

“My method must use tools” (C).

“The reflexology stick protects both the client and the reflexology practitioner” (D).

Similarly, respondents (B, F, G, H, and J) also expressed that a massage stick is used (Table 5).

The interview survey discovered that 59.3% of reflexologists would use gadgets when appropriate (Table 2). However, 25 reflexologists practice Fr. Josef Eugster’s reflexology health method and use tools. Respondents in this group will use the white foot reflexology stick (FJM stick) developed by the organization to assist in applying force and in preventing occupational injuries (Figure 4).

4. What do you think about the reflexology license?

“Great, at least a professional affirmation” (B).

“We use the Ministry of Health and Welfare to issue them a certificate; yes, only Taiwan has this system” (C).

“We have to test and change the license every year, so the method does not regress” (E).

“Our reflexology association must take the exam for the annual internal examination, and I will also find a way to take the government’s massage examination. After all, one more license provides more security” (J).

## 4. Discussion

### 4.1. Key Elements and Suggestions for Prevention of Musculoskeletal Injuries by Reflexologists

Jang et al. (2006) studied visually impaired massage practitioners and found a 71.4% incidence of work-related musculoskeletal disorders [41]. Later, research by Chang (2010) reported that 64.2% of masseuses had physical discomfort [42]. The incidence of hand osteoarthritis symptoms among female masseuses was first studied by Kruger et al. (2017). They observed that 60% of the subjects had pain in their hands, and 64% of the subjects felt stiffness in their hands the next day [43]. According to the research results of Wu (2015), the primary services of beauticians include oil pressure, finger pressure, skincare, makeup, etc. More than one musculoskeletal discomfort instance in the past 12 months was reported by 89.3% of the respondents [22]. Busa et al. (2017) found musculoskeletal wrist disorders over the past 12 months in 74.14% of chiropractors [44]. According to Lee et al. (2017), 89.5% of respondents who performed massages for the visually impaired in Taiwan reported musculoskeletal discomfort in the past 12 months [45]. Sirbu et al. (2022) found an 88.09% prevalence of work-related musculoskeletal discomfort among massage practitioners [46]. They observed that compared to other massage-related personnel, foot reflexology therapists had a lower rate of musculoskeletal discomfort at only 33.9%, which may be because foot reflexology therapists perform warm-up exercises, small tool assistance, and mutual massage health care. In this study, 79.7% of foot reflexology therapists reported performing self-protection exercises such as deep breathing, mutual massage, and warm-up before work; thus, basic work protection and warm-up exercises can promote body flexibility and confer injury prevention benefits [47]. To apply force to the reaction zone, 59.3% of foot reflexology therapists use massage sticks to assist respondents. According to research, reflex sticks can help foot reflexology therapists easily find reflex points on the feet of specific individuals [48] and reduce unnecessary effort. At least once a week, 42.4% of foot reflexology therapists perform massages using a mutual massage maintenance mechanism. Reflexology can relieve the severity of musculoskeletal pain in certain parts of the body of caregivers and reduce fatigue [49,50]. A weekly mutual care regimen with a foot reflexology therapist may help foot reflexology practitioners reduce their musculoskeletal discomfort and allow immediate repair. All of the above findings may be reasons for the lower prevalence of musculoskeletal injuries among foot reflexology therapists. Among the respondents, 57.6% reported that their work chairs are fixed without backrests and pulleys, and ergonomics cannot be used to improve the design of reflexology products. Lower workbenches place substantial pressure on the lower back and cause a 3.6 times increase in the probability of lower back pain and discomfort [24,41]. Choosing an adjustable bench height allows the podiatrist to maintain a correct posture for longer working hours, and the bench is appropriate if the foot reflexology therapist’s body weight can be used as an additional aid [51]. Taifa and Desai (2017) also recommend designing adjustable facilities wherever possible, as work equipment design can have a significant impact on reducing the occurrence of musculoskeletal injuries [52]. In addition, Buck et al. (2007) showed that using a massage table involves a higher degree of use of the lumbar erector spinae of the foot reflex therapist, resulting in a significant increase in mild trunk flexion; the anterior muscles are used more heavily, causing significantly more severe radial wrist deviation and mild shoulder flexion [24]. Therefore, a mechanism to adjust the height of the workbench can protect the spine of the foot reflexology therapist in the correct working position [53].

### 4.2. Assessment of the Impact of PES on Reflexology Practitioners

The survey found that foot reflexology therapists had high psychological empowerment scores, which is strong support for the profession of reflexology and indicates that reflexology therapists have a high degree of autonomy in their work in terms of implementing positive psychological empowerment (PE) relative to organizational and personal performance [54]. Specifically, PE partially mediated the relationship between work engagement [55]. PE enhances job performance [56]. It also shows that the foot reflexologist affirms the importance of reflexology in life. Correspondingly, it is implied that the reflexology specialist has a high level of confidence in performing the job. Respondents had the lowest average score on the organizational impact of reflexology work. PES was developed by Spreitzer [40]. PES is measured by four dimensions: meaning, competence, self-determination, and impact [57]. Among the four dimensions, the meaning of work and self-efficacy scored the highest, while the influence dimension scored the lowest. The top four scoring items are as follows: “On the meaning dimension, what I do in my work is very meaningful to me personally. On the talent dimension, I have mastered the skills needed to get the job done. I have the confidence to do everything well at work. I am confident in my ability to get the job done.” These facilitate foot reflexology therapists in following their own ideas and standards to judge the value of work, hold the belief of having enough skills for good performance, and have independent advocacy and moderate action choices, which will affect the organization’s strategy, administrative management, and operating results and promote self-fulfillment of the foot reflexology teacher [58]. The top three lowest-scoring dimensions are as follows: “I have a large influence on what happens in this department, and I have a large influence on what happens in this department on the impact dimension. It’s a good control and I have a major influence on what happens in this department.” However, because most foot reflexology therapists in Taiwan are hired by stores and often work part-time, their administrative influence on work organization is therefore less significant. Future research can develop a better cooperation system between the store owner and the foot reflexology therapist (employee) so that the therapist can participate in decision making, development, and organization operations and achieve a win–win situation. Specifically, psychological empowerment partially mediated the relationship between work engagement.

### 4.3. Taiwan’s Clinical Foot Reflexology Therapists’ Cognition and Current Situation of Complementary Medicine Therapy

Foot therapy practitioners have a positive attitude towards foot therapy, believing it is a self-serving natural therapy that benefits others. Research has shown that reflexology maintains and promotes health and not only treats disease [59]. Foot therapy can improve physical function, overall health, and wellbeing [60,61]. Foot reflexology practitioners advocate that foot therapy is different from foot massage. Foot massage relieves soreness by rubbing body parts. In contrast, foot reflexology is a self-healing technique that involves pressing specific pressure points on the sole and back of the foot [62]. Foot reflexology is the most accurate name [63]. The interviewed foot reflexology practitioners reflected a BMI that is slightly higher than the BMI of 24.7 in Taiwan’s labor anthropometric results [64], and higher BMI (overweight and obesity) values have been linked to the increased prevalence of musculoskeletal symptoms [65]. Most foot reflexology practitioners are in good health. Research reveals that 57.5% of Chinese visually impaired massage workers are unhealthy [66]. In this study, only 8.5% felt that their health was poor, and 68% felt that they were in good health. The data also show that most of foot reflexology practitioners feel happy in their living conditions and have high job satisfaction with foot reflexology. This study found that those in the foot reflexology occupation have a long employment life paired with suitable occupations. According to Blau’s (2012) study, it was discovered that the job of a foot massager negatively correlated with work fatigue and physical exhaustion, but this was not observed in this study. This demonstrates that the weekly working hours of a reflexologist are approximately 36.5 h, which is consistent with Taiwan’s legal working hours of 40 h per week [67]. The monthly salary of a reflexologist could be between TWD 40,000–60,000. Compared with the average monthly salary of TWD 32,539 in the general service industry in Taiwan [68], the salary is approximately 1.8 times that of other service industries, which may contribute to job satisfaction. The most frequently served customers are those with chronic diseases, and the top three include insomnia, muscle ache problems, and gastrointestinal problems (Figure 3). The findings are broadly consistent with the effects of insomnia [15,16], colic [6] , low back pain [7], and joint pain [8]. Both reflexologists and professional associations advocate reflexology therapy as effective for general health maintenance and treating chronic diseases such as stroke, musculoskeletal disorders, and stress [69]. Additionally, foot therapy positively affects the digestive system, autonomic nervous system, skin, physical symptoms, nerve sensation, and urinary incontinence [63]. As shown in Figure 2, the proportion of female foot reflexology therapists is the highest among the 45–59-year-old group of foot reflexology therapists. There are many new female residents in Changbin, eastern Taiwan, as they married far away from Taiwan [70]; their behavior is highly restricted by their husband’s families and social status [71]. With the help of Fr. Josef Eugster H.D. Assoc., learning the foot reflex health method helps restore health and provides a job to help these people attain economic independence. Moreover, colleagues in the workplace support each other, providing emotional comfort, suggesting that the foot therapy industry is suitable for both sexes.

### 4.4. The Attitude of Foot Reflexology Therapist towards Examination

Similarly to other countries, foot massage legality falls into a gray area; it is only officially legal when provided by a medically qualified professional; otherwise, it can only coexist with a formally regulated healthcare system or is not involved within the scope of internal activities related to medical efficacy [72]. To solve this legally controversial issue, in 2015, the government of Taiwan included foot massage as a folk conditioning industry [73] and set its tone as a health care function rather a medical treatment. This study found that foot reflexology therapists are positive and willing to cooperate with the government’s plan to establish an assessment system for foot reflexology therapists. In a survey, 64–85% of the public agreed that it is necessary for folk recuperation practitioners to receive a formal education, training, and qualification certification [74]. Embong et al. (2017) reported that formal training is not required because reflexology is not yet legally recognized [75]. Not taking continuing education classes can put foot reflexology therapists at greater risk of injury. Hence, for all foot reflexology therapists, it is important to continue these strengthening exercises over time to prevent musculoskeletal discomfort [76]. This study thus appeals to the relevant government units in terms of establishing a professional certification system for foot reflexology therapists as soon as possible, affirming professional foot reflexology therapists in meeting public expectations, and also maintaining the safety of foot reflexology therapists at the workplace and the rights and interests of customers.

### 4.5. Limitations

These findings, similarly to all studies, cannot be generalized to a larger population, and the opinions expressed in this study do not represent the experience of all Taiwanese reflexologists. Due to the impact of the COVID-19 epidemic, Taiwan’s foot reflexology therapists were forced to close down their workplaces in 2021. For example, the FJM work platform in Changbin, eastern Taiwan, closed for 2 months (17 May 2021~27 July 2021). During the period of this study, therapists had just returned to work, while some have not even resumed working. Hence, it was difficult to conduct face-to-face interviewees. This study can be used as an example (after epidemic measures are relieved) to obtain more samples and to further confirm the findings in this study. The survey and interview should be timed as closely as possible at the convenience of the reflexologist, avoiding the period before and after Chinese New Year.

### 4.6. Recommendations

This article discusses occupational injuries and psychological empowerment among foot reflexology therapists in Taiwan. Healthcare practices that should be beneficial to the general public have led to occupational harm for foot reflexology therapists. This is indeed a question worthy of reflection, and further research can be performed.

## 5. Conclusions

This research concludes that the following:In total, 33.9% of reflexologists have symptoms of musculoskeletal discomfort in body parts, with the left shoulder (25.4%), left hand or wrist (25.4%), lower back (25.4%), right shoulder (23.7%), and left elbow or forearm (22%) being the top five most uncomfortable areas. Moreover, the average pain level was between no pain and slight pain (0.2–0.6).Reflexology practitioners are highly psychologically empowered to work.Foot therapy practitioners have a positive attitude towards foot therapy, believing it as a self-serving natural therapy that benefits others.Most reflexologists support the government’s desire to promote the reflexology examination system and are willing to cooperate with the policy’s development.The height of most work chairs does not match the height of the guest seat and is not ergonomic, and this can be used as a reference for improving the design of foot reflexology products.The foot reflexologist job offers middle-aged women good job skills, a sense of achievement, and financial ability.There are different factions of foot therapists. Fr. Josef Eugster’s foot reflexology health faction requires the use of tools in treatments to penetrate deep into the reflexology zone and uses the lever principle to save effort in order to protect the muscles and bones of the foot therapist.

## Figures and Tables

**Figure 1 healthcare-11-00394-f001:**
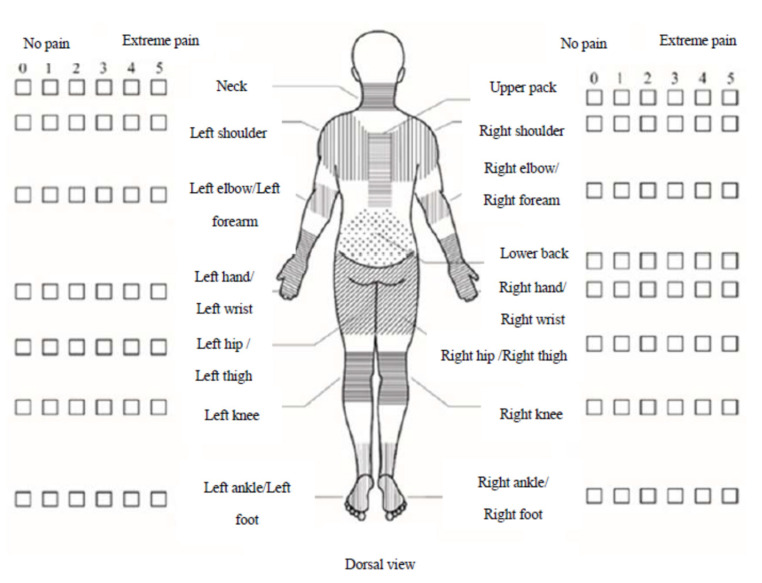
Modified Nordic Scale.

**Figure 2 healthcare-11-00394-f002:**
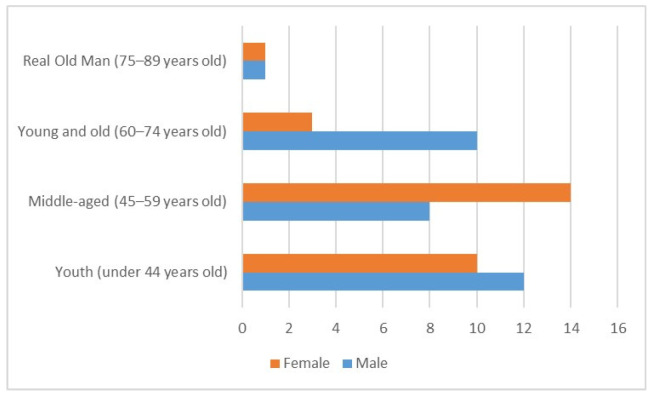
Age distribution of men and women.

**Figure 3 healthcare-11-00394-f003:**
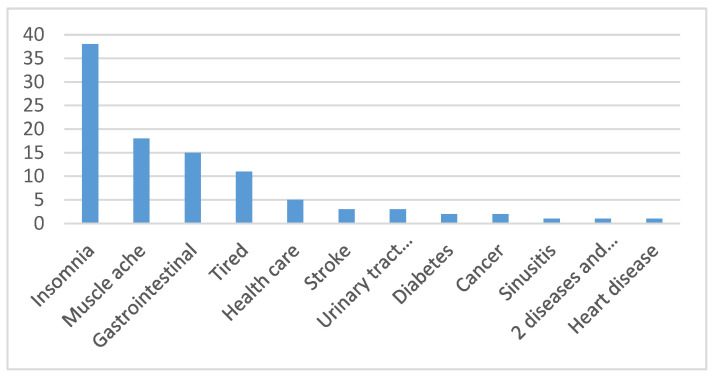
The most common cases of foot therapy in practice.

**Figure 4 healthcare-11-00394-f004:**
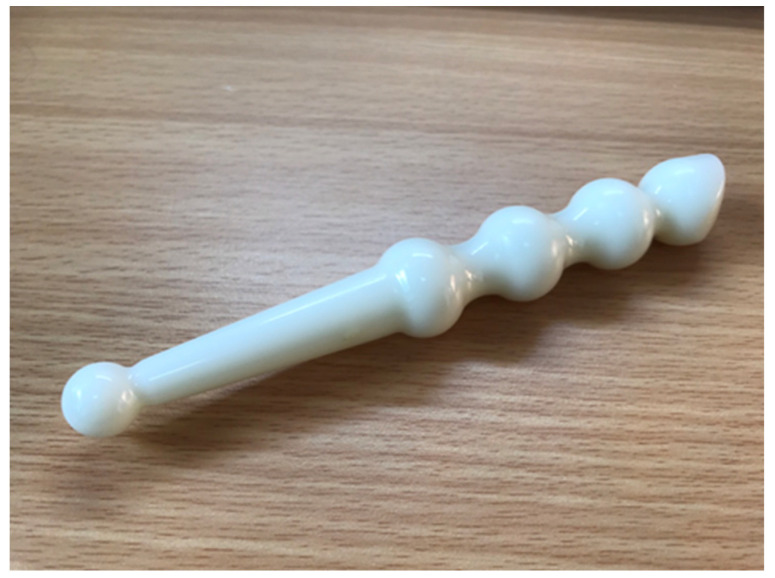
FJM rods used in plantar reflexology.

**Table 1 healthcare-11-00394-t001:** Statistical table of basic information of the respondents.

Project	N	Min	Max	Mean	SD
Basic information					
Age	59	22	81	49.0	14.0
Height	59	150	182	165.1	8.8
Weight	59	46	105.0	68.4	12.2
Living condition					
Perceived health status	59	1	5	3.9	1.0
Happiness in life	59	1	5	4.0	1.0
Job satisfaction	59	1	5	4.1	1.0
Working conditions					
Working days per week	59	0.5	7	5.2	1.6
Daily working hours	59	0.1	12	7.3	4.0
Number of customers served per day	59	1	10	3.8	1.7
Degree of love for work	59	1	5	4.1	1.0
Willingness to study	59	1	5	4.0	1.1
Work environment satisfaction	59	1	5	4.0	1.0
Seniority (years)	59	0.1	36	9.6	9.5

**Table 2 healthcare-11-00394-t002:** Relevant survey results of reflexologists.

Project	Frequency	Percentage		Cumulative Percentage
**Educational level**				
College graduate	22	37.3		37.3
High school graduate	1	35.6		72.9
Elementary school Bi	9	15.3		88.2
Junior high school graduate	7	11.9		100
**Total**	**59**	**100**		
**Other vocational training**				
None	38	64.4		64.4
Beauty salon related	6	10.2		74.6
Chef	5	8.5		83.1
Massage	3	5.1		88.2
Maintenance vehicle	2	3.4		91.6
Agricultural training	1	1.7		93.3
Accountant training	1	1.7		95
Lifeguard training	1	1.7		96.7
Carpentry training	1	1.7		98.4
Life orientation	1	1.7		100
**Total**	**59**	**100**		
**Previous occupation**				
None	26	44.1		44.1
Business	19	32.2		76.3
Labor	7	11.9		88.1
Clerk	2	3.4		91.5
Agriculture	3	5.1		96.6
Others	2	3.4		100.0
**Total**	**59**	**100**		
**Disease**				
None	45	76.3		76.3
Eye disease	4	6.8		83.1
Hypertension Diabetes	3 3	5.1 5.1		88.2 93.3
Heart disease	1	1.7		95.0
Bronchial disease	1	1.7		96.7
Neck pain	1	1.7		98.4
Ear disease	1	1.7		100
**Total**	**59**	**100**		
**Trauma**				
None	35	59.3		59.3
Spine	6	10.2		69.5
Foot	5	8.5		78.0
Hand	4	6.8		84.8
Head	2	3.4		88.2
Neck and shoulders	2	3.4		91.6
Knee	2	3.4		95.0
Eye	1	1.7		96.7
Teeth	1	1.7		98.4
Face	1	1.7		100
**Total**	**59**	**100**		
**Perceived health status**				
Very unhealthy	1	1.7		1.7
Not very healthy	4	6.8		8.5
Average	14	23.7		32.2
Healthy	21	35.6		67.8
Very healthy	19	32.2		100
**Total**	**59**	**100**		
**Leisure**				
None	25	42.4		42.4
Walk	15	25.4		67.8
Volunteer	6	10.2		78.0
Travel	4	6.8		84.8
Gym	3	5.1		89.9
Climb mountains	2	3.4		93.3
Swim	1	1.7		95.0
Ball	1	1.7		96.7
Run	1	1.7		98.4
Bike	1	1.7		100
**Total**	**59**	**100**		
**Nature of the work**				
Full-time	47	79.7		79.7
Part-time	12	20.3		100
**Total**	**59**	**100**		
**Workplace**				
Work platform	53	89.8		89.8
To the house	3	5.1		94.9
Studio + home	2	3.4	_	98.3
Studio	1	1.7		100
**Total**	**59**	**100**		
**The biggest trouble**				
None Affected by the epidemic	34 8	57.6 13.6		57.6 71.2
Personal problem	5	8.5		79.7
Question from guest	4	6.8		86.4
Work problem	8	13.6		100
**Total**	**59**	**100**		
**Self-protection exercises**				
None Deep breathing	12 6	20.3 10.2		20.3 30.5
Mutual massage	24	40.7		71.2
Warm-up exercises	17	28.8		100
**Total**	**59**	**100**		
**Use devices**				
None	24	40.7		40.7
Yes	35	59.3		100
**Total**	**59**	**100**		
**With designed chairs or other working chairs**				
Other With designed chair	34 25	57.6 42.4		57.6 100
**Total**	**59**	**100**		
**Work chair height**				
20 cm 28 cm	1 11	1.7 18.6		1.7 20.3
30 cm	3	5.1		25.4
31 cm	3	5.1		30.5
34 cm	5	8.5		39.0
38 cm	2	3.4		42.4
38 to 48.5 cm	25	42.4		84.7
40 cm	4	6.8		91.5
44 cm	2	3.4		94.9
70 cm	1	1.7		96.6
90 cm	2	3.4		100
**Total**	**59**	**100**		
**Served seat with designed chair**				
Other With designed chair	34 25	57.6 42.4		57.6 100
**Total**	**59**	**100**		
**Served Seat Height**				
120 cm 30 cm	2 1	3.4 1.7		3.4 5.1
41 cm	6	10.2		15.3
42 cm	1	1.7		16.9
43 cm	7	11.9		28.8
45 cm	2	3.4		32.2
47 cm	5	8.5		40.7
50 cm	7	11.9		52.5
55 cm	2	3.4		55.9
58 to 64 cm	25	42.4		98.3
85 cm	1	1.7		100
**Total**	**59**	**100**		
**Served person poses**				
**Sitting**	**59**	**100**	**100**	
Musculoskeletal discomfort				
None	39	66.1		66.1
Yes	20	33.9		100
**Total**	**59**	**100**		

**Table 3 healthcare-11-00394-t003:** Musculoskeletal status of reflexologists.

Project	N	Min	Max	Mean	SD	Number of Patients	Percentage
Neck	59	0	3	0.3	0.8	1 0	16.9
Left shoulder	59	0	3	0.5	1.0	15	25.4
Left elbow/left forearm	59	0	4	0.4	0.9	13	22.0
Left hand/left wrist	59	0	4	0.6	1.1	15	25.4
Left hip/left thigh	59	0	4	0.3	0.8	7	11.9
Left knee	59	0	3	0.2	0.7	6	10.2
Left ankle/left foot	59	0	4	0.2	0.7	6	10.2
Upper back	59	0	4	0.4	1.0	12	20.3
Right shoulder	59	0	4	0.4	0.9	14	23.7
Right elbow/right forearm	59	0	3	0.3	0.7	12	20.3
Lower back	59	0	4	0.5	1.1	15	25.4
Right hand/right wrist	59	0	5	0.5	1.2	12	20.3
Right hip/right thigh	59	0	4	0.3	0.9	9	15.3
Right knee	59	0	3	0.3	0.7	8	13.6
Right ankle/right foot	25	0	3	0.2	0.6	6	10.2

**Table 4 healthcare-11-00394-t004:** The work attitude of reflexologists.

	N	Min	Max	Mean	SD
Meaning of work					
1. Work makes sense	59	3	5	4.4	0.7
2. Personally meaningful	59	2	5	4.5	0.7
3. Importance of work	59	3	5	4.2	0.8
Autonomy					
4. Decide on work	59	2	5	4.4	0.7
5. Independence	59	2	5	4.4	0.8
6. Autonomy	59	2	5	4.3	0.8
Talent					
7. Master the skills	59	3	5	4.5	0.6
8. Confidence	59	3	5	4.5	0.6
9. Ability Confidence	59	3	5	4.5	0.6
Work impact					
10. Affect the degree of work	59	1	5	3.2	1.1
11. Control	59	1	5	3.2	1.1
12. Influence of the individual on organization	59	1	5	3.1	1.1

**Table 5 healthcare-11-00394-t005:** Survey results related to the interview of the reflexologists.

Project	Frequency	Percentage	Cumulative Percentage
**Opinion on foot reflexology**
Improve health	10	100	100
Total	100	100	
Help people Other	64	6040	60100
Total	6	100	
Increase economic incomeOther	28	2080	20100
**Total**	**10**	**100**	
**Sense of accomplishment**
Improves the disease	10	100	100
**Total**	**10**	**100**	
Curing the disease	5	50	50
Other	5	50	100
**Total**	**10**	**100**	
**Use reflexology tools**
Yes	10	100	100
**Total**	**10**	**100**	
**Reflexology license**
Job requirements	7	70	70
Professional recognition	2	20	90
Policy cooperation	1	10	100
**Total**	**10**	**100**

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
