# Peer review of "Investigation on Musculoskeletal Injury and Psychological Empowerment of Reflexologists in Taiwan: Analysis of the Recognition to Alternative Therapy"

_healthcare, 2023, doi:10.3390/healthcare11030394_

Round 1
Reviewer 1 Report
- The article describe a positive process of regulating the reflexology in Taiwan, helping avoiding non professional reflexologist.
- The novelty of the article is nit's dealing with the practitioners themselves rather trying tp prove once again the efficacy of reflexology.
- The results are applicable to the practitioners themselves concerning agronomic chair
- The article is well written (both English and methodology wise), clear, easy to understand.
Author Response
請參考附件

Reviewer 2 Report
Looking at the reflexologists’ working conditions is very interesting and important. However, form a scientific perspective, significant improvement is needed.
Abstract
Confusing, and does not provide a clear description of the aim, method, and result.
For example
Interviews were used in the study, but it is not mentioned in the abstract. However, results from the interviews are mentioned in the result section.
The aim is slightly different from the aim in the manuscript
Introduction
The introduction tells a lot about foot reflexology and less about the reason for the study. It says few studies in Taiwan have discussed… What do these studies tell about working as a reflexologist
Materials and Methods
The following information is missing
- - how the informants were recruited – applies to both the questionnaire and interview.
- - how many questionnaires were sent and how many were answered (the response rate)
- - how was the questionnaire validated
- - the rationale for the choice of questions – applies to both the questionnaire and interview
In general, if 1159 are working as a reflexologist, 59 responses are very few.
Ethical considerations are also missing. For example approval and consent to participate.
An analysis section is missing. How was the interview analyzed?
No reference to support the authors’ choices etc.
Results
In general, the number of results/information is overwhelming. Despite this the number of information the section is not very informative.
Discussion
No discussion of the result, moreover no discussion of the result in relation to other studies, exemplified by just one reference.
The section limitation is difficult to assess due to the lack of information in the section: materials and methods.
Reviewer 3 Report
This paper reports on a survey of reflexologists in Taiwan. We have reviewed the submitted manuscript and comment on the following points.
1.(line.78) Please describe the selection criteria for the 59 reported respondents. We believe that the results of this survey can only be discussed by adding information on what population they represent.
2.(line.84) Height and weight are listed among the survey items. What is the purpose of this information? Please consider whether or not information that is not mentioned in the results of the survey should be included. For example, if there is a possible relationship with musculoskeletal pain, please consider the relationship and discuss the content.
3.(Title)The title "Musculoskeletal injury" is only included in the questions of the questionnaire, but there is no description about the relationship with other survey items. Musculoskeletal injury may be related to working environment as well as length of service, gender, and age. You need to cite the literature as appropriate and provide evidence that the effect of work environment is significant.
4. Similar to 3, in Psychological Empowerment, it is necessary to consider what factors bring about Empowerment from the results.
5. (Table5)In Table 5, there is "Survey results related to the podiatrist interview". If it is a survey of reflexologists and not podiatrists, the wording needs to be changed.
Round 2
Reviewer 2 Report
Thank you for the opportunity to read an improved version of the article.
Abstract: Improved, as it now to a greater extent describes the study.
Introduction: The background and the purpose of the study are clearer.
Materials and methods: The section has been improved, yet further improvements are needed. Alternatively, the deficiencies need to be mentioned in the section: limitations.
Materials and Methods: My last comment: How was the questionnaire validated? - A question related to face validity, content validity. I still do not find any information about this.
Materials and Methods: An analysis section is still missing. You refer to p. 3 line 132 -141. Line that provides information on the questions, but no information on analysis.
A section should provide information such as descriptive analysis and the definition of effective percentage (table 2). What is the difference between effective percentage and percentage? Asking as the data in the two columns is the same.
Same results is an example of overwhelming and not very informative information, which I find is still the case.
With regards to the interview and analysis; it is mentioned that it is developed by two experts (line 134), yet no information on the background for these questions, and still no information on how data is analyzed. The latter would also include reasoning for the use of citations as well as a table /table 5. A table that is not completely coherent with using semi-structured interviews.
Materials and Methods: No reference to support authors’ choices. A comment regarding the choice of e.g. Nordic Scale. A reference has been added, but no reference regarding PES (p. 12 line 301), moreover what does PES stands for?
It is good, that several references have been added, but they are often used for the discussion of the results, instead of supporting the choices.
The discussion is now supported by the use of other studies/references, which is good, but as mentioned the results are also discussed in the result-section. The discussions should be gathered in one place; in the discussion.
Reviewer 3 Report
Correction of the indicated points has been confirmed.
